# Effects of degraded speech processing and binaural unmasking investigated using functional near-infrared spectroscopy (fNIRS)

**Xin Zhou**[1]*, **Gabriel S. Sobczak**[2], **Colette M. McKay**[3,4], **Ruth Y. Litovsky**[1,5,6]

1 Waisman Center, University of Wisconsin-Madison, Madison, WI, United States of America, 2 School of Medicine and Public Health, University of Wisconsin-Madison, Madison, WI, United States of America, 3 The Bionics Institute of Australia, Melbourne, VIC, Australia, 4 Department of Medical Bionics, University of Melbourne, Melbourne, VIC, Australia, 5 Department of Communication Science and Disorders, University of Wisconsin-Madison, Madison, WI, United States of America, 6 Division of Otolaryngology, Department of Surgery, University of Wisconsin-Madison, Madison, WI, United States of America

* xzhou353@wisc.edu

**Data Availability Statement:** All relevant data are within the paper and its Supporting Information files.

## Abstract

The present study aimed to investigate the effects of degraded speech perception and binaural unmasking using functional near-infrared spectroscopy (fNIRS). Normal hearing listeners were tested when attending to unprocessed or vocoded speech, presented to the left ear at two speech-to-noise ratios (SNRs). Additionally, by comparing monaural versus diotic masker noise, we measured binaural unmasking. Our primary research question was whether the prefrontal cortex and temporal cortex responded differently to varying listening configurations. Our *a priori* regions of interest (ROIs) were located at the left dorsolateral prefrontal cortex (DLPFC) and auditory cortex (AC). The left DLPFC has been reported to be involved in attentional processes when listening to degraded speech and in spatial hearing processing, while the AC has been reported to be sensitive to speech intelligibility. Comparisons of cortical activity between these two ROIs revealed significantly different fNIRS response patterns. Further, we showed a significant and positive correlation between self-reported task difficulty levels and fNIRS responses in the DLPFC, with a negative but non-significant correlation for the left AC, suggesting that the two ROIs played different roles in effortful speech perception. Our secondary question was whether activity within three sub-regions of the lateral PFC (LPFC) including the DLPFC was differentially affected by varying speech-noise configurations. We found significant effects of spectral degradation and SNR, and significant differences in fNIRS response amplitudes between the three regions, but no significant interaction between ROI and speech type, or between ROI and SNR. When attending to speech with monaural and diotic noises, participants reported the latter conditions being easier; however, no significant main effect of masker condition on cortical activity was observed. For cortical responses in the LPFC, a significant interaction between SNR and masker condition was observed. These findings suggest that binaural unmasking affects cortical activity through improving speech reception threshold in noise, rather than by reducing effort exerted.

**Funding:** This study was supported by National Institute on Deafness and Other Communication Disorders (NIH-NIDCD, R01DC003083 to RL), UW-Madison's Office of the Vice Chancellor for Research, and a Core grant from NIH-NICHD (U54HD090256 to Waisman Center).

**Competing interests:** Dr. Litovsky discloses that she is a consultant for Frequency Therapeutics. This does not alter our adherence to PLOS ONE policies on sharing data and materials. The other authors have all certified that they have NO affiliations with or involvement in any organization or entity with any financial interest or non-financial interest in the subject matter or materials discussed in this manuscript.

## Introduction

Listening to speech can be challenging in many situations, for instance, when communicating in a "cocktail party" scenario whereby the listener is surrounded by multiple conversations [1]. Limitations also arise when individuals with hearing loss need to extract information from acoustically degraded speech [2]. Listeners may expend elevated cognitive resources in a challenging condition to retain accuracy in speech perception [3]. On a long-term basis, chronically-elevated cognitive resources while listening to sounds in the environment could result in fatigue, decreased quality of life and reduced work efficiency [4]. Effortful speech processing is closely tied to auditory cortex (AC) activity involved in sound perception, and to lateral prefrontal cortex (LPFC) activity associated with higher-level speech understanding. However, as reviewed below, studies to date report different sub-regions of the LPFC responding to degraded speech. Conflicting evidence also exists with regards to LPFC activity in response to different spatial configurations of noise relative to the target speech. The current study focused on binaural unmasking, whereby speech understanding can improve when target speech is spatially separated from masking noise compared to when the target and maskers are co-located [5–9]. We were primarily interested in whether functional near-infrared spectroscopy (fNIRS) could reveal differences in cortical activities between the left LPFC and AC in the context of binaural unmasking at different speech-to-noise ratios (SNRs). Our secondary question was whether there were functional differences across sub-regions within the LPFC in varying stimulus configurations.

### fNIRS measures of cortical activity

The fNIRS imaging method uses near-infrared light that travels through the superficial cortical areas, with some of the light photons being absorbed by the chromophores and some being scattered [10]. By measuring changes in light intensity as a function of time, fNIRS reveals the concentration changes of oxygenated and deoxygenated hemoglobin, denoted as $\Delta$HbO and $\Delta$HbR respectively, in the local cortical area contained within the pathway of the infrared light. The concentration changes in hemoglobin (called hemodynamic responses) are thought to be closely related to the neuronal activity in the cerebral tissue through neural vascular coupling [11]. Previous studies have shown that fNIRS measures are closely related to the blood-oxygen-level-dependent (BOLD) signal from functional magnetic resonance imaging (fMRI), with strong correlations shown between BOLD signals and $\Delta$HbO [12], between BOLD and $\Delta$HbR [13, 14], and between BOLD signals and the total concentration changes in hemoglobin, i.e., $\Delta$HbT [15]. The differences between $\Delta$HbO and $\Delta$HbR, i.e., $\Delta$HbO—$\Delta$HbR, which assess changes in cerebral oxygenation ($\Delta$HbC), have also been used to reveal changes in neuronal activity in the prefrontal cortex [16–18]. One of the advantages of fNIRS over fMRI is the compatibility with ferromagnetic materials (e.g., metal implants), thus fNIRS is optimal for measuring cortical activity in populations where fMRI scanning is precluded [19]. Further, compared to the loud scanning noise of fMRI, fNIRS is silent and has been popular for research involving auditory stimuli [20–26].

### Neuroimaging studies revealed evidence of binaural unmasking

The current study was designed to garner evidence for binaural unmasking based on cortical activity measured using fNIRS. To induce binaural unmasking, we presented monaural speech stimuli to the left ear with noise presented either monaural (co-located) or diotic (both ears with no interaural difference in time or intensity). These configurations were tested with speech that was either unprocessed or vocoded. When comparing the monaural and diotic noise conditions, improved performance in speech reception thresholds is indicative of

binaural unmasking, which facilitates source segregation and improved speech intelligibility in noisy environments [5–9]. Neural representations of binaural unmasking have been recorded in the brainstem, mainly in the inferior colliculus [27, 28] and in the AC of mammals [29]. In humans, previous studies investigated the neural origins of binaural unmasking using techniques such as brainstem auditory evoked potentials, frequency-following responses, and auditory steady-state responses. However, these studies did not demonstrate evidence of binaural unmasking at the brainstem level [see ref. 30 for reviews]. The absence of observed subcortical contributions to binaural unmasking in humans could be due to the limited spatial resolution of the neuroimaging systems. It could also be due to some unknown mechanisms that are likely specific to humans and need further exploration. At the cortical level, while some studies demonstrated effects of binaural unmasking in auditory areas [31, 32], other studies did not [33, 34]. The inconsistencies of results could be due to differences between studies in the stimuli and task conditions that were used. Interestingly, neural correlates of binaural unmasking were reported [35], with greater N1 amplitude in the unmasked versus masked condition in NH adults and in adults with atrophy in the brainstem including the inferior colliculus. This result suggested that in humans, effects of binaural unmasking persist at the cortical level even with severe damage in the brainstem.

A previous fMRI study [36] investigated the neural correlates of binaural unmasking by presenting speech and noise in conditions with diotic stimuli in two ears, or with phase-inverted speech or noise in one ear (dichotic). They found an area in the left inferior frontal gyrus (IFG) that showed significant differences in BOLD signals when speech or noise was phase-inverted in one ear, compared to in the diotic condition. A recent fNIRS study [37] focused on the role of auditory spatial attention while listening to speech with an informational masker in which speech and noise were either spatially separated or co-located. They found that two areas of the lateral frontal cortex (LFCx) showed significantly greater activity when target speech and masker were spatially separated versus co-located. The authors interpreted their findings to suggest that both sides of the LFCx on both hemispheres could be involved in spatial attention processing and binaural unmasking, both of which can contribute to improving speech perception. To summarize, as both the IFG and LFCx in the above two studies [36, 37] were within the LPFC, effects of binaural unmasking in humans have been associated with changes in cortical activity in the LPFC on both sides and in the AC.

## LPFC and AC contribute differently to speech perception

The bilateral AC and left LPFC are closely connected when auditory perception, phonological and semantic processing are considered [see 38 for meta-analyses, 39]. However, these brain regions have been reported to show different response patterns in speech perception experiments. A neuroimaging study using positron emission tomography (PET) investigated speech perception of sentences in noise [40]. As the level of masking noise decreased, speech intelligibility improved, as expected; the left anterior AC showed increased responses and left IFG showed decreased responses in the regional cerebral blood flow (rCBF). A previous fMRI study [41] investigated cortical activity for auditory identification ('ba' versus 'da') with varying levels of masking noise. The results showed that, as the masking noise level decreased, the identification accuracy increased, and was positively correlated with bilateral AC activity. The reaction times in identifying the targets were longer at a medium level of masking noises [41], and shorter when the task was hard or very easy. Further, reaction times were positively correlated with activity in the left IFG (anterior insula-operculum), suggesting that left IFG was associated with task demands hence possibly varying listening effort. Another study on this topic using fNIRS [42] examined speech perception in noise in naturalistic scenes and found

elevated cortical activity in the right dorsolateral prefrontal cortex (DLPFC) as SNR decreased, task demand increased and speech was reported as being more difficult to understand.

Different effects of spectral degradation on cortical activity of the AC and LPFC have also been observed when listeners performed auditory perception tasks. For instance, a previous fNIRS study [23] examined cortical responses for spectrally degraded speech at varying levels of intelligibility (0, 25, 50, 75, and 100%). They found that, within the range of conditions when the speech was intelligible (between 25% and 100% correct), as the intelligibility increased, responses in the ACs increased and responses in the left IFG decreased. Their results in the AC replicate previous findings showing that magnitude of cortical responses in the AC are sensitive to, and positively correlated with speech intelligibility [40, 43–45]. When speech intelligibility was reduced (but above zero) and the task demands were higher, increased cortical activity in the left IFG likely reflected greater cognitive resources being exerted to understand speech in more challenging conditions. Similarly, an increased activity in the left IFG when attending to degraded speech versus unprocessed speech has been reported in previous studies using fMRI [46, 47] and fNIRS [20]. The AC was activated regardless of attention directed towards speech or distractors, suggesting that attention did not significantly alter cortical responses in the AC as compared to the LPFC [20, 46].

Results from above studies together seem to suggest that the left AC is sensitive to SNRs and spectral degradation. The LPFC has quite different response patterns to AC, and attention plays an important role in modulating activity in this region.

## Sub-regions of the LPFC

Previous studies were in agreement that the LPFC is involved in effortful speech perception in challenging conditions, though with mixed results regarding which subregions of the LPFC are involved. For instance, the DLPFC on the left [36] and right [37] seem to be involved in attentional listening to speech in noise and spatial processing, with hemispherical differences possibly related to differences in experimental configurations. The left IFG seems to be involved in effortful perception of speech as speech intelligibility decreases, either due to spectral degradation or masking noise. The differences in regions involved could be partially due to the varying number of optodes employed in different fNIRS montages across experiments which have impacted the size of recording region (surface area). For instance, 4 channels on the LFCx in this fNIRS study [37] permitted a greater recording area compared with the 3 channels on the IFG in another study [20]. Without a good coverage of the frontal area, it is difficult to parse out whether LPFC subregions overlap and share common functions, and whether the regions reported in the fNIRS studies [20, 23, 37] overlap with the regions reported in fMRI studies [36, 46, 47]. Alternatively, different subregions within the LPFC might contribute differently to binaural unmasking and the processing of spectrally degraded speech in noises.

## Goals of the current study

The current study investigated the effects of spectrally degrading speech and binaural unmasking on cortical activity measured using fNIRS. Our primary interest was in fNIRS measures in the DLPFC and AC. Because of the unclear roles of LPFC subregions, our secondary research question was whether fNIRS measures could reveal the differences among the subregions within the LPFC with varying stimulus configurations. Besides the DLPFC, we also examined two adjacent regions of interest (ROIs) within the LPFC, which corresponded to Brodmann area (BA, 9 and 46), and BA45 and BA47 on the surface. Besides fNIRS measures, this study also assessed subjective assessments of task difficulty and accuracy of speech intelligibility.

We predicted that, due to binaural unmasking, listeners would report listening to speech with diotic noise to be easier than with monaural noise. In addition, we predicted that unprocessed speech would be easier than vocoded speech. We hypothesized that when task demands were higher but not impossible to perform, and assuming that listeners remained motivated, more cognitive resources would be spent [3, 48] manifested as greater fNIRS responses in the LPFC, compared to when task demands were lower. For our primary research question, based on previous research, we predicted that fNIRS responses in the AC would show an opposite trend compared with the response pattern in the DLPFC across conditions. For our secondary research question, we predicted that if varying configurations, i.e., spectral degradation, binaural unmasking, and noise level, had different effects on each sub-region in the LPFC, we would see different patterns between the three subregions.

## Methods

### Participants

Twenty-seven volunteers were recruited for the fNIRS session. Four volunteers were excluded at the beginning as 'acceptable' light intensity for fNIRS data collection was not obtained due to hair artifacts. Twenty-three adults advanced to the testing phase (13 women; mean and standard deviation (SD) of ages: 22.7 ± 3.1 years, range 19–30 years; 21 right-handed). These participants were recruited through a university-run online job posting site at the University of Wisconsin-Madison and were paid for their time. A different group of 15 volunteers (13 women, mean ± SD: 21.7 ± 2.8 years, range 19–29 years) were recruited to participate in a separate behavioral speech perception task. These were undergraduate students at the University of Wisconsin-Madison and participated in the study for credits. All the participants were native English speakers (none were bilingual) with normal pure tone thresholds at or less than 20 dB HL with less than 10 dB difference between two ears at octave frequencies between 125 Hz and 8000 Hz. Experimental protocols were within standards set by the National Institutes of Health and approved by the University of Wisconsin–Madison's Human Subjects Institutional Review Board. All participants provided written consent.

### Stimuli

The speech stimuli consisted of matrix sentences each having the same structure with monosyllabic words from 5 categories: name, verb, number, adjective, object, with 8 words in each category [49]. An example sentence is: '*Bob sold six blue socks.*' For each sentence, in each of the 5 categories of words, one of 8 options was randomly chosen, thereby creating grammatically correct but unpredictable sentences. Both unprocessed (U) and vocoded (V) versions of the sentences were used. The speech was vocoded using a white-noise carrier whereby the spectrum was divided into eight frequency bands between 200 Hz and 7000 Hz, i.e., 8-channel noise-vocoded [2], with filters based on Greenwood functions. The noise stimuli were 8-channel noise-vocoded 4-talker babble of IEEE sentences [50]. Both the matrix sentences and the IEEE sentences were recorded by American woman speakers. The speech (S) stimuli were delivered to the left ears, i.e., monaurally (m) through an ER-2A insert earphone (Etymotic). Two noise (N) configurations were implemented. In the first configuration, noise was presented to the ear with the speech stimulus, i.e., monaurally at 60 dBA ($F_{max}$, maximum level with A-weighted frequency response and Fast time constant), referred to as $N_m S_m$. In the second configuration, the noise was presented to both ears, i.e., diotically, at 57 dBA ($F_{max}$) with no interaural time differences, referred to as $N_o S_m$. The 3-dBA reduction in the $N_o S_m$ condition was introduced to compensate for the otherwise doubling of intensity, so that the sound pressure level would be equalized between $N_m S_m$ and $N_o S_m$ conditions. Speech was presented

at two SNRs, -15 and -10 dB. The same noise stimuli were presented in the unprocessed and vocoded sentence conditions.

## fNIRS data collection and signal processing

**fNIRS system and experimental montage.** The fNIRS system used in this study (NIRScout, NIRX medical technologies, LLC) was a continuous-wave NIRS instrument with 16 LED light sources (Fig 1A, red dots) and 16 avalanche photodiode (APD) detectors (Fig 1A, blue dots). Each LED light source emitted near-infrared light with wavelengths of 760 nm and 850 nm. A source with each of its adjacent detectors at 3 cm constituted measurement channels (Fig 1A, yellow lines). A NIRScap (NIRX medical technologies, LLC) was used to hold the light sources and detectors on the head.

Because fNIRS signals of interest (neuronal activity-related changes in hemoglobin from cerebral tissue) are contaminated by responses from the extracerebral tissue, such as systemic and non-evoked brain responses [52], it is essential to reduce such confounds and improve the neural signal quality. In the current study, a bundle of 8 detectors (Fig 1A, 4 green dots on each side) were used, which were situated 8 mm from the light sources, providing "short channels" [53]. The short channel photon path was shallow and expected to only reveal responses in the superficial extracerebral tissue but not the cerebral tissue [54]. Regressing out the short-channel components from the regular fNIRS channels has been shown to improve fNIRS signal quality [53, 55–58], i.e., the ratio of cerebral to extracerebral components. fNIRS responses were examined in three sub-regions in the LPFC and AC on both hemispheres. Each sub-region consisted of three channels (Fig 1B). The DLPFC corresponded to the Broadman area (BA, 9 and 10) on the surface, f-ROI2 corresponded to BA 9 and 46, and f-ROI3 corresponded to BA45 and BA47, which likely covers the IFG. Fig 1B plots the sensitivity profile of each region on the left side [59]. The sensitivity profiles were generated with AtlasViewer software [51] and revealed the light intensity changes over the given area underneath channels in each region.

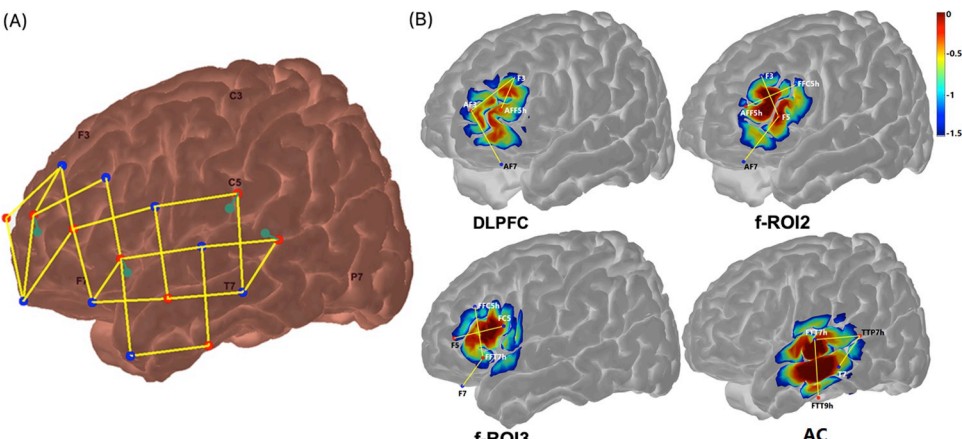

**Fig 1. fNIRS montage and a priori regions of interest (ROIs).** The fNIRS montage was symmetric between hemispheres; panel (A) plots the connection of light sources (red, n = 8) and detectors (blue, n = 8), and channels (yellow lines) on the left hemisphere. The green dots denote detectors that provide 8-mm channels, with 4 on each side. Panel (B) shows channels comprising the three subregions within the left lateral prefrontal cortex (LPFC) and the left auditory cortex (AC). The three regions were the dorsolateral prefrontal cortex (DLPFC), and two adjacent regions of interest, i.e., f-ROI2 and f-ROI3. The colors are the sensitivity profiles, in log10 mm-1 units, generated from AtlasViewer [51].

**Data collection.** The fNIRS data were collected in a standard IAC sound-attenuated booth, with participants sitting in an armchair wearing a NIRSCap of predetermined size for a snug fit around the head. During preparation, to centralize the cap and to correctly position the optodes, the Cz was positioned halfway the distance of Nasion to Inion and halfway the distance between the two pre-auricular points. Further, the frontal optodes Fp1 was positioned at 10% of the Nasion-Inion distance (a few centimeters above the eyebrows). The cap was attached to a chest wrap for fixation. Then the gains of light intensity at the APD detectors were checked to ensure that all the channels had at least 'acceptable' light intensity. If some channels did not show good intensity, the most likely factors were either that the optodes were not perpendicular to the scalp or hair strands interfering with the photon path. To rectify this problem, the optodes were taken out and the hair underneath was gently pushed away to create better contact with the skin before replacing the optodes. The optimization procedure was repeated until most of the optodes received at least acceptable light intensity. Four out of 27 participants with less than half of the channels showing acceptable light intensity were excluded from the study with no further fNIRS data being collected.

A pseudo-random block design was implemented for fNIRS data collection consisting of six 9-minute testing periods. Each testing period (Fig 2B) started with a 30-second silent period for baseline data collection, followed by a block of stimuli from one of the four listening

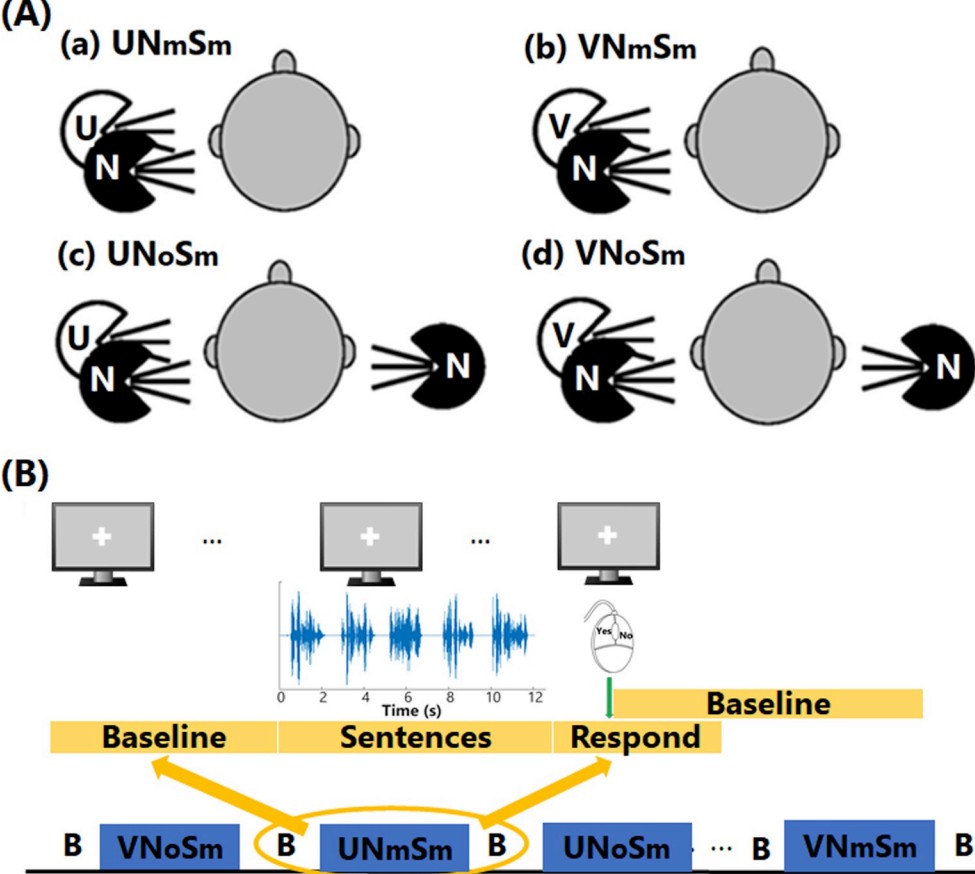

**Fig 2. Listening conditions and diagram of fNIRS data collection.** In panel (A), the white loudspeakers are for unprocessed (U) and vocoded (V) speech; the black loudspeakers are for noise (N), monaurally ($N_mS_m$) or diotically ($N_oS_m$) presented. Panel (B) shows the pseudorandom block design used for data collection, with stimuli in 4 listening conditions (blue boxes) being presented in random order in each session.

conditions, i.e., unprocessed or vocoded speech, with either monaural noise on the speech side or diotic noise (see Fig 2A) at one SNR (-10 or -15 dB). In each testing period, 3 blocks per condition were presented in random order. Among the six testing periods, the order of the 2 SNRs was randomized. Each block lasted 13.6 s and consisted of 5 sentences, with a 0.6-s interval between sentences. After each block, there was a jittered silent period (25 to 35 s in duration). In total, nine blocks of stimuli per condition were presented. The experiment was run in the Presentation® software (https://www.neurobs.com/), which is a stimulus delivery and experiment control platform.

Participants were required to attend to the speech stimuli by counting the number of color words in each block and then to click a mouse button to respond immediately after the block was finished. Participants were instructed to click the left or right buttons, when the number of recognized color words was even or odd, respectively, and to click the middle button (scroll wheel) if they did not understand any of the words. A practice session prior to fNIRS testing was conducted with each participant, to familiarize them with varying configurations of speech, the pattern of the fNIRS data collection, and the task. During practice, they listened to the vocoded speech in quiet and in noise. A block design was used with varying lengths of silence between blocks and participants were required to do the same task as described above. Verbal instruction was given by the experimenter; text instruction was available either on a brochure or a monitor 1.5 m in front of the participants throughout the testing.

**fNIRS signal processing.** The fNIRS signals recorded by the NIRScout system were imported to MATLAB (MATLAB R2017a) for further analysis, with software that was either written by the authors or using scripts adopted from Homer2 [60]. A short-channel subtraction method was applied by extracting the principal component in the eight 8-mm channels, and regressing these out from the regular fNIRS channels to reduce the extracerebral components in the fNIRS data [58]. The steps of signal processing were as follows (see Fig 3).

**1) Remove step-like noise.** Step-like noise can be caused by a sudden loss of contact between optodes and the skin, or interposition of hair, during data collection. To remove step-like artifacts in the data (y) of each channel, the deviation of y was first estimated as $X = \text{diff}(y)$. Any absolute values in $X$ that were two SD above the mean of $X$ were set as zeros, i.e., if $\text{abs}(X_i) > \text{mean}(\text{abs}(X)) + 2^*\text{std}(X)$, then $X_i = 0$. Response $y$ (with step-like artifacts removed) was then recovered by calculating the cumulative sum of the updated $X$, i.e., $y_{\text{post}} = \text{cumsum}(X)$.

**2) Exclude 'poor' channels.** Channels of 'poor' data quality should be excluded from further analysis. As heartbeat signals are the salient signals in the fNIRS measurements, channels that fail to record the heartbeat signals are unlikely to record other physiological or neural responses. To quantify the heartbeat signals, the correlations between heartbeat signals (0.5–1.5 Hz) in the intensity data of two different NIR wavelengths [61], i.e., the scalp coupling index (SCI), were calculated. In the current study, the cut-off SCI threshold was set as 0.15, with the same threshold used in [53], to ensure for each participant there were at least 4 out of 8 short channels remaining for further analysis [58]. Our previous study [53] also demonstrated that using the threshold of SCI >= 0.15 and SCI >= 0.75 that was recommended [61], resulted in comparable signal qualities after short-channel subtraction, measured as contrast-to-noise ratios. Further, keeping SCI >= 0.15 ensured that short channels from both the frontal and temporal cortex that measured extracerebral responses from both ROIs, were involved in further analysis. The mean ± SD ratios of regular channels and short channels that were excluded were 2.45% ± 4.46% and 11.05% ± 14.75%, respectively.

**3) Preprocess and calculate the ΔHbO and ΔHbR.** Light intensity data in individual channels were first converted to optical density [60]. A wavelet decomposition method proposed in [62] was then performed to correct motion artifacts, which might be caused by the physical displacement of the optodes from the surface of the participant's head. With wavelet

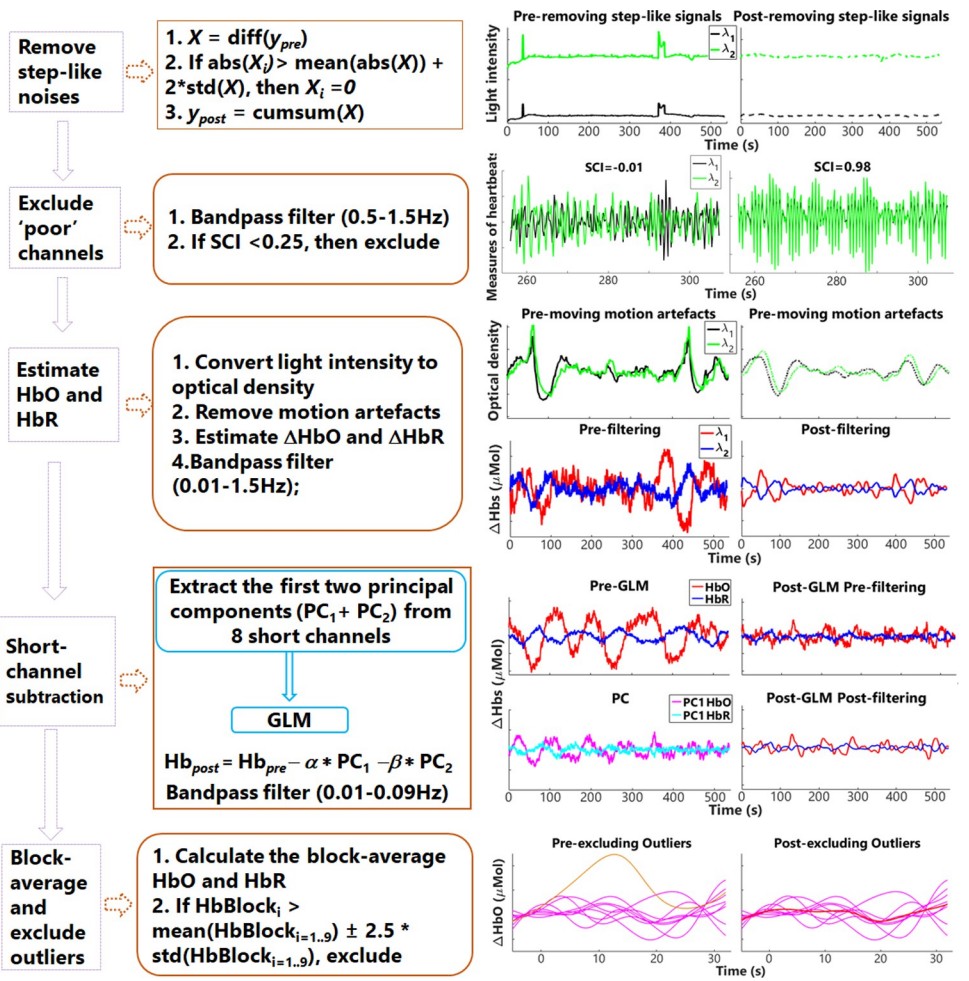

**Fig 3. Diagram of fNIRS signal processing.**

decomposition, motion artifacts appear as abrupt breaks in the wavelet domain, whereas hemodynamic responses to stimuli have fewer variable coefficients. To remove the motion artifacts, wavelet coefficients above 0.1 interquartile were set to zero, the same setting as in [21]. Finally, the concentration changes of ΔHbO and ΔHbR responses were calculated using the modified Beer-Lambert law [63], with the effect of age and wavelengths of near-infrared light on the calculation of differential pathlength factor adjusted [64].

**4) Subtract the short-channel component.** A principal component analysis (PCA) was performed on HbO and HbR responses separately from short channels with SCI $> = 0.15$. The mean ± SD of short channels involved in PCA among participants was 6.91 ± 1.33. The first two principal components (PCs) among all that contributed the most to the short-channel responses were assumed to be the 'global' components across channels and needed to be removed. The mean ± SD of the total variances that two PCs contributed to the HbO and HbR responses were, 77.25% ± 6.39% and 73.21% ± 10.43%, respectively. The two PCs were used as regressors in a general linear model (GLM), the product of which and the corresponding coefficients from GLM were then subtracted from HbO and HbR signals, separately, in each channel. A third-order Butterworth band-pass filter (cut-off frequency at 0.01–0.09 Hz) was

applied to remove the high-frequency physiological signals [65], such as respiration and heartbeats.

**5) Average responses across blocks and exclude outliers.** Block-average responses were calculated for ΔHbO, ΔHbR and ΔHbC, with baseline averages of each block, i.e., 5 s before stimulus onset, being subtracted. All the blocks were inlcuded, regardless of participants' accuracies in pushing a mouse button to indicate hearing an even or odd number of color words, except for individual blocks that had values above or below the mean ± 2.5 SD of the group. The means of block-average responses across channels that clustered into ROIs were calculated.

**6) Quantify ΔHbC responses.** Further analyses were performed on ΔHbC amplitudes, which combined ΔHbO and ΔHbR information and were calculated by first identifying the peak of the responses within 5–17 s after stimulus onset. The means within 5 s of ΔHbC responses centered at the peaks were then calculated for individual channels for each participant in each condition.

**7) Calculate ΔHbC amplitudes in ROIs.** For our primary research interest, ΔHbC amplitudes channels located above the frontal (n = 10) and temporal (n = 10) regions on each hemisphere were first averaged, separately. Channels above the LPFC were further clustered into three subregions based on their anatomic positions. Including AC, fNIRS responses were examined in four ROIs on both hemispheres, with each ROI consisting of three channels.

## Scoring self-reported task difficulty

Alongside fNIRS data, subjective assessment of the task difficulty levels was acquired from all participants. Immediately after each of the 6 testing periods, participants were asked to score the difficulty in understanding the sentences (on a scale of 0 to 10, with 0 corresponding to no difficulty and 10 corresponding to extremely difficult; for details, see Fig 4A). For each participant, the self-reported difficulty score for each condition consisted of the mean of the 3 difficulty scores measured from the three testing periods.

## Scoring speech intelligibility

To evaluate the average effect of listening conditions on speech intelligibility, a different group of participants who had not been exposed to the stimuli before were tested with no fNIRS data being collected. Participants listened to a set of matrix sentences and responded to one sentence at a time by using a computer mouse to click on the buttons, with a closed set of words displayed on a monitor in the front. There was a break after each sentence and the task was performed at the individual's own pace. After every 5 sentences in the same condition, participants scaled the task difficulty from 0 to 10 through a computer program. The order of listening conditions was randomized and a total of 20 sentences (100 words) per condition were tested. The accuracy was calculated as the percentage of correct responses participants made per condition, then a rationalized arcsine transform [66] was used to transform accuracy into speech intelligibility. Note that the speech intelligibility task was different from the color word identification task performed during fNIRS data collection, with the latter requiring participants to count the number of color words in 5 sentences but not to identify each word. The color-word counting task was designed to keep listeners' attention to the stimuli while avoiding frequently pushing buttons or articulating during the speech presentation, which would result in motion artifacts and motor cortical activity contaminating fNIRS data.

## Statistical analysis

Statistical analyses were carried out using R (R Core Team, 2019). Aligned rank transform (ART) tests [67], which are nonparametric factorial analyses of variance (ANOVA), were

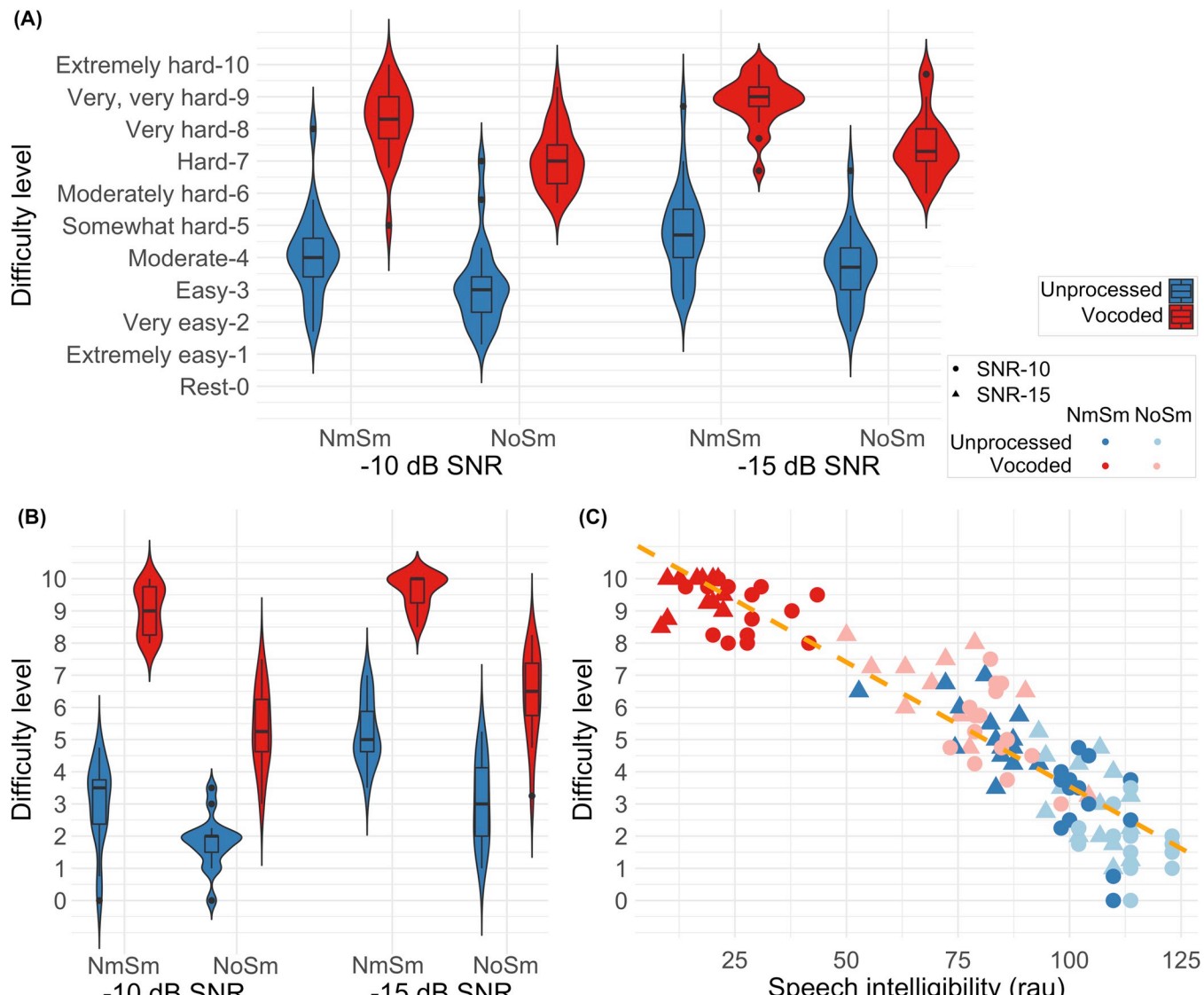

**Fig 4. Subjects' self-reported difficulty levels and speech intelligibility scores.** Unprocessed (U, blue) or degraded (V, red) speech stimuli were always presented to the left ear alone. Noise stimuli were presented in ipsilateral ($N_mS_m$) or bilateral ($N_oS_m$, squares) conditions. In panel (A), violin plots show self-reported difficulty levels in individuals under varying listening conditions at -10dB SNR (left) and -15 dB SNR (right) during fNIRS recording. Panel (B) shows the self-reported difficulty levels in a separate group of participants in a behavioral session with no fNIRS data being recorded. Panel (C) shows the correlation between the self-reported difficulty levels (vertical) and speech intelligibility in rationalized arcsine units (RAU, horizontal) in individuals in the behavioral session.

conducted for two reasons. First, the self-reported difficulty scores were ordinal; second, fNIRS data were not normally distributed and variances were not spherical. Thus, data were subjected to nonparametric statistical tests. ART tests ('ARTool' package) with a mixed model ('lmer' package) were conducted separately on 1) the self-reported difficulty levels during the fNIRS session, 2) difficulty levels from the separate behavioral test, and 3) the speech intelligibility results, with speech type (unprocessed or vocoded), masker condition ($N_mS_m$ or $N_oS_m$), and SNR (-10 dB or -15 dB) as fixed factors and participant as a random factor. *Post hoc* pairwise comparisons within single factors were conducted using estimated marginal means ('emmeans' package) and Tukey method for p-value adjustment. The function

'testInteractions' ('phia' package) was used for significant interactions between factors. Repeated measures correlations [rmcorr, 68] were calculated between the speech intelligibility scores and difficulty levels from the behavioral test (without fNIRS data collection).

For fNIRS measures, we first conducted an ART test to compare ΔHbC amplitudes between the frontal and temporal regions on both hemispheres to confirm the regional differences in cortical activity during speech perception. To address our primary research question whether the left DLPFC and AC responded differently to speech with varying configurations, we conducted an ART test to compare fNIRS measures between these two regions, with speech type, masker condition, SNR, and ROI as fixed factors and participant as a random factor. For *post hoc* analyses, the same methods were used as for the behavioral measures. Further, to examine fNIRS measures of task demands, repeated measure correlations were calculated between self-reported task difficulty level for varying conditions (n = 8) and ΔHbC amplitudes in the left DLPFC and AC.

To address our secondary question whether the three sub-regions within the LPFC responded differently, we conducted another ART test. As the first ART test identified a significant difference between two hemispheres, with greater activity on the right than on the left, we hence examined LPFC on both hemispheres. The ART test was conducted with speech type, masker condition, SNR, ROI and hemisphere as fixed factors and participant as a random factor.

## Results

### Self-reported task difficulty levels during fNIRS testing

Fig 4A shows results from the subjective assessment of task difficulty in individuals during fNIRS recording and the rating scale that was used. At both SNRs, participants reported lower difficulty for the unprocessed speech (in blue) versus for the vocoded speech (in red), and lower difficulty for speech in the left ear with diotic noise ($N_oS_m$) versus with monaural noise ($N_mS_m$), suggesting that binaural unmasking may have reduced task difficulty. The vocoded speech with monaural noise was judged to be the hardest condition, with a few participants reporting this condition being extremely hard at -15 dB SNR. Results from ART tests, as reported in Table 1, found significant main effects of SNR, speech type, and masker condition ($N_mS_m$ versus $N_oS_m$), all with $p < 0.001$. The results revealed increased difficulty in understanding vocoded speech compared to unprocessed speech, and less difficulty in listening to speech with diotic noise compared to with monaural noise. Results from ART tests also found a significant interaction between the speech type and the binaural unmasking ($F(1,154) = 4.08$, $p = 0.045$), with greater effect of binaural unmasking in the vocoded condition than in the unprocessed condition.

**Table 1. Summary of results from ART tests for behavioral measures.** The behavioral measures are task difficulty level (TDL) recorded during fNIRS session and in the behavioral session with no fNIRS, and speech intelligibility scores (SIC) recorded in the behavioral session.

| | SNR | Speech | Masker | SNR * speech | SNR * masker | Speech * masker | SNR* speech * masker |
|---|---|---|---|---|---|---|---|
| TDL (fNIRS) | F(1,154) = 24.83 *p* < .001 | F(1,154) = 669.25 *p* < .001 | F(1,154) = 90.96 *p* < .001 | F(1,154) = 0.78 *p* = .38 | F(1,154) = 0.11 *p* = .74 | **F(1,154) = 4.08 *p* = .045** | F(1,154) = 0.75 *p* = .39 |
| TDL (no fNIRS) | **F(1,98) = 66.69 *p* < .001** | **F(1,98) = 485.30 *p* < .001** | **F(1,98) = 258.44 *p* < .001** | **F(1,98) = 5.61 *p* = .020** | F(1,154) = 0.36 *p* = .55 | **F(1,98) = 27.09 *p* < .001** | **F(1,98) = 4.26 *p* = .042** |
| SIC (no fNIRS) | **F(1,98) = 153.77 *p* < .001** | **F(1,98) = 509.01 *p* < .001** | **F(1,98) = 494.08 *p* < .001** | **F(1,98) = 5.21 *p* = .025** | **F(1,98) = 7.26 *p* = .008** | **F(1,98) = 268.99 *p* < .001** | **F(1,98) = 5.02 *p* = .027** |

## Behavioral results without fNIRS recording

Fig 4B plots the self-reported difficulty levels from a separate group of 15 individuals in a behavioral session without fNIRS recording. As shown in Fig 4B, the patterns of self-reported task difficulty levels across listening conditions in the group involved in behavioral tasks were similar to that in the other group who reported the difficulty levels during the fNIRS session (Fig 4A). Results from ART tests, as reported in Table 1, also showed significant main effects of SNR, speech type, and masker condition on this set of self-reported task difficulty levels, all with $p < 0.001$. Results also found a significant interaction between SNR, speech type, and masker condition (F(1, 98) = 4.26, $p = 0.042$).

Fig 4C plots the self-reported difficulty levels versus speech intelligibility scores in each condition at two SNRs. Results from ART tests on the speech intelligibility scores showed significant main effects of SNR, speech type, and masker condition (Table 1). Results also showed significant interactions between the following factors: SNRs * masker condition, SNRs * speech type, speech type * masker condition, and SNR * speech type * masker condition. Further, results from the repeated measure correlation analysis found a significant correlation between self-reported task difficulty levels and speech intelligibility scores ($r = -0.95$, $p < 0.001$), suggesting that speech intelligibility decreased as listeners reported the tasks being more difficult.

## fNIRS responses in the LPFC and AC

**Frontal versus temporal cortex.** Fig 5 plots the group means (markers) and SEMs (error bars) of ΔHbC amplitudes for the frontal (orange) and temporal cortex (green) on the left and right hemispheres. For each region, the ΔHbC amplitudes across 10 channels were first averaged for individuals. As shown in Fig 5 and from an ART test, ΔHbC amplitudes in the frontal cortex on both hemispheres were greater compared to in the temporal cortex ($t(710) = 6.42$, $p < 0.001$), with greater responses on the right hemisphere compared to the left ($t(710) = 2.04$, $p = 0.042$). Results from an ART test did not find a significant interaction between cortical regions and hemispheres ($F(1,710) = 1.11$, $p = 0.29$). The significantly greater amplitudes on the right versus left hemisphere, were likely because speech stimuli were always presented in the left ears, which might result in greater contralateral than ipsilateral cortical activity.

**Comparing responses in the left DLPFC versus AC.** For our primary research interest, we focused on fNIRS measures from the left DLPFC and AC. Fig 6 shows the group mean (markers) and SEM (bars) of the ΔHbC amplitudes for the left DLPFC (panel A) and the AC (panel C) for unprocessed (blue, circles) and vocoded (red, triangles) speech with diotic ($N_oS_m$) and monaural ($N_mS_m$) noise. For each ROI, results for -10 dB and -15 dB SNR were plotted on the left (solid lines) and right (dash lines) columns. As shown in Fig 6, the left DLPFC and AC showed opposite patterns across conditions. Results from the ART test found a significant difference between the two ROIs, with smaller responses in the left DLPFC than the AC (see Table 2). Further, there were significant interactions between speech type * ROI, and between SNR * ROI, with the left DLPFC showing greater differences between vocoded and unprocessed conditions and greater responses at -15 dB versus -10 dB SNR, compared to the left AC. Fig 6B shows the repeated measure correlation results between self-reported task difficulty level and the ΔHbC amplitudes in the two ROIs. Results demonstrated a significant and positive correlation for the left DLPFC (panel B; $r = 0.266$, $p = 0.004$), suggesting a neural marker in the left DLPFC for task demands, with a negative but non-significant correlation for the left AC (panel D; $r = -0.134$, $p = 0.09$). The effect size in the DLPFC was relatively small ($r = 0.266$). Indeed, the correlation was driven by the larger responses to -15 dB versus -10 dB SNR, and greater responses to vocoded versus unprocessed speech, but not binaural

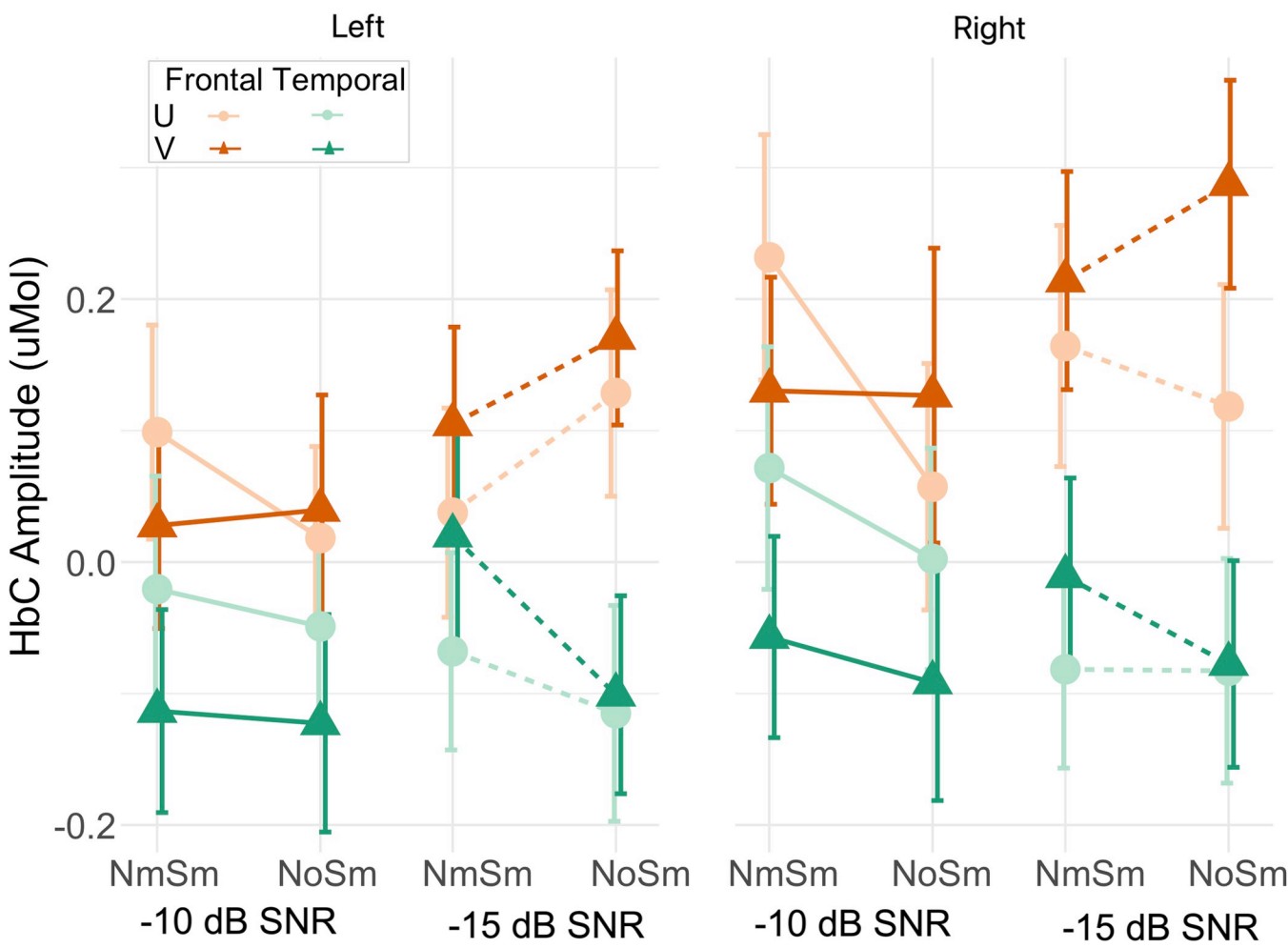

**Fig 5. ΔHbC amplitudes in the frontal and temporal cortex.** Group mean ΔHbC amplitudes for the frontal (orange) and temporal (green) cortex on the left and right hemispheres in response to unprocessed (dots) and vocoded speech (triangles) at -10 dB (solid lines) and -15 dB SNR (dash lines) are plotted.

unmasking. We did not observe smaller responses in the $N_oS_m$ conditions, which were self-reported as easier compared to the $N_mS_m$ conditions.

**Responses in the three subregions in the LPFC.** To address our secondary question, the fNIRS responses were examined in the three sub-regions within the LPFC on both hemispheres. Fig 7 plots the group mean (markers) and SEMs (shaded errors) of ΔHbC amplitudes on the left and right hemispheres, for the DLPFC (panels A, B), f-ROI2 (panels C, D) and f-ROI3 (panels E, F). An ART test was conducted on the ΔHbC amplitudes for the three subregions on the two hemispheres. Results showed a significant effect of speech type, SNR, ROI and hemisphere, and a significant interaction between masker condition * SNR. Detailed results are reported in Table 2. *Post hoc* analyses found greater responses to the vocoded versus unprocessed speech ($t(1034) = 1.97$, $p = 0.049$) and greater responses on the right hemisphere compared to the left ($t(1034) = 3.46$, $p < 0.001$). Between three ROIs, f-ROI2 showed greater ΔHbC amplitudes compared to the DLPFC ($t(1034) = 3.12$, $p = 0.005$) and f-ROI3 ($t(1034) = 3.92$, $p < 0.001$). For the interaction between binaural unmasking and SNR, *post hoc* analysis found greater differences between $N_oS_m$ and $N_mS_m$, i.e., masker condition, at -15 dB SNR compared to -10 dB SNR ($\chi^2(1) = 5.60$, $p = 0.018$).

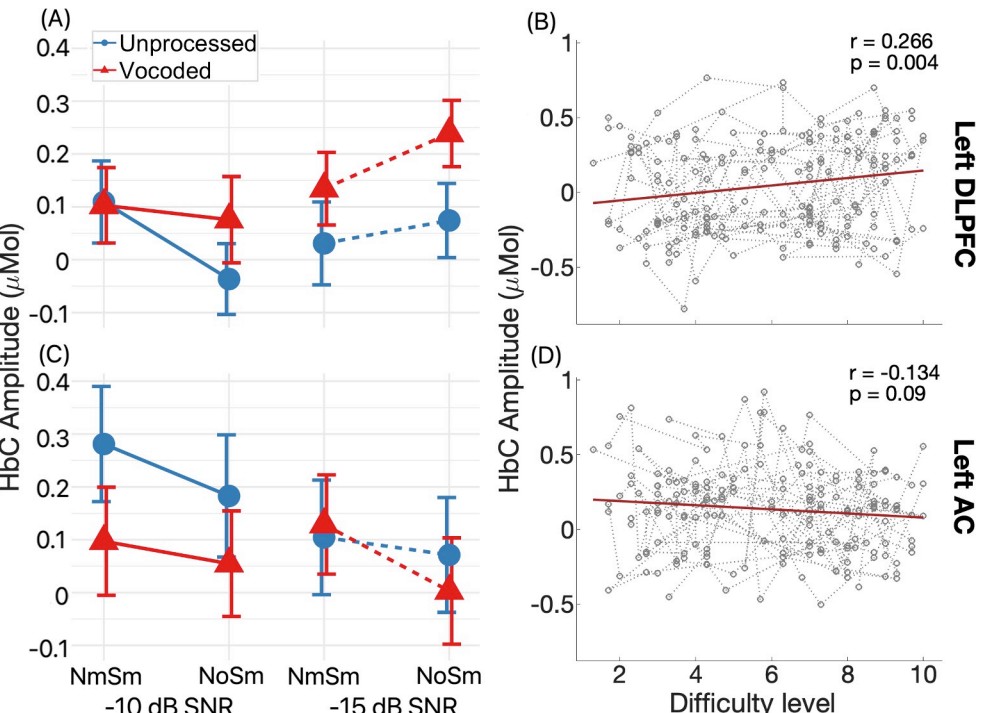

**Fig 6. ΔHbC amplitudes in the left DLPFC and AC.** Panels A and C show the group mean (bars) and SEM (error bars) of ΔHbC amplitudes in the left DLPFC and AC, respectively, in the unprocessed (blue dots) and vocoded (red triangles) speech conditions with monaural ($N_mS_m$) and diotic noise ($N_oS_m$) at -10 dB (solid lines) and -15 dB SNR (dash lines). Panels (B) and (D) show the repeated measures correlations (rmcorr) between ΔHbC amplitudes in the left DLPFC and AC, respectively, and self-reported task difficulty level. In each panel, the gray dash lines connecting circles represent measures in individuals in different conditions; the red lines indicate the regression result from the rmcorr method.

## Discussion

In the current study, we used fNIRS to investigate cortical activity in response to vocoded versus unprocessed speech, and to compare conditions with monaural versus diotic noises at two different SNRs. These configurations were selected to understand how the brain responds when listening to speech with varying configurations designed to induce binaural unmasking. To ascertain task demands and participants' performance, respectively, we also recorded participants' self-reported task difficulty and speech intelligibility for each condition. For fNIRS responses, we were primarily interested in the LPFC and the AC as the two regions have been reported to respond differently depending on attention and speech type. Our secondary interest was whether each of the three sub-regions within the LPFC was more sensitive to some configurations than the others.

### LPFC and AC responded differently to two speech types

We expected greater cognitive resources would be spent *to overcome obstacles in goal pursuit* in the more challenging conditions [3]. This would be manifested as greater changes in the ΔHbC (cerebral oxygenation) amplitudes, which are associated with increased neuronal activity through neurovascular coupling [11]. Our results showed significantly greater changes in the ΔHbC amplitude in the frontal compared to the temporal regions when responding to varying types of configurations. Further, our *a priori* analysis found different response patterns of ΔHbC amplitudes between the left DLPFC and AC, with the left DLPFC showing greater

**Table 2. Summary of results from ART tests for fNIRS measures.** The left and right sides summarize the results related to our primary and secondary research questions, relatively. We investigated the effect of speech type by comparing unprocessed (U) versus vocoded (V) speech, and the effect of masker condition by comparing diotic ($N_oS_m$) and monaural ($N_mS_m$) conditions, the effect of SNRs (-10 and -15 dB SNR), and the interactions between them on different cortical regions.

| Frontal versus temporal regions | | | DLPFC, f-ROI2, and f-ROI3 | | |
|---|---|---|---|---|---|
| Factor(s) | ART results | Post hoc | Factor(s) | ART results | Post hoc |
| Region | **F(1, 710) = 41.22 p < .001** | **frontal > temporal t(710) = 6.42, p < .001** | Hemisphere | **F(1, 1034) = 11.96 p < 0.001** | **Right > left t(1034) = 3.46, p < .001** |
| Hemisphere | **F(1, 710) = 4.14 p = .042** | **Right > left t(710) = 2.04, p = .042** | ROI | **F(1, 1034) = 8.58 p < 0.001** | **f-ROI2 > DLPFC, p = .005 f-ROI2 > f-ROI3, p< .001** |
| Hemisphere*region | F(1, 710) = 1.11 p = .29 | | Hemisphere*ROI | F(1, 1034) = 2.75 p = 0.064 | **DLPFG > f-ROI3: Right—Left** |
| **Left DLPFC versus AC** | | | SNR | **F(1, 1034) = 5.08 p = 0.024** | **-15dB > -10dB SNR t(1034) = 2.25, p = .024** |
| Factor(s) | ART results | Post hoc | Speech | **F(1, 1034) = 3.89 p = 0.049** | **V > U t(1034) = 1.97, p = .049** |
| ROI | **F(1,330) = 6.61 p = .011** | **DLPFC < AC t(330) = 2.57, p = .011** | SNR*speech | F(1, 1034) = 3.35 p = 0.068 | -15 dB SNR > -10dB SNR V vs U |
| ROI*SNR | **F(1,330) = 5.30 p = .022** | **LPFC > AC: -15 vs -10dB SNR** | Masker*SNR | **F(1, 1034) = 5.60 p = 0.018** | **NoSm vs NmSm: -15dB > -10 dB SNR** |
| ROI*speech | **F(1,330) = 6.54 p = .011** | **LPFC > AC: V vs U** | Masker | F(1, 1034) = 0.13 p = 0.72 | |
| Masker | F(1,330) = .12 p = .72 | | Masker*hemisphere | F(1, 1034) = 2.11 p = 0.15 | |
| SNR | F(1,330) = .15 p = .69 | | Masker*ROI | F(1, 1034) = 0.039 p = 0.67 | |
| Speech | F(1,330) = .20 p = .65 | | Masker*speech | F(1, 1034) = 0.79 p = 0.38 | |
| Masker*ROI | F(1,330) = .019 p = .89 | | Hemisphere*SNR | F(1, 1034) = 0.062 p = 0.80 | |
| Masker*SNR | F(1,330) = .003 p = .96 | | Hemisphere *speech | F(1, 1034) = 0.077 p = 0.78 | |
| Masker*speech | F(1,330) = 1.56 p = .21 | | ROI*SNR | F(1, 1034) = 0.063 p = 0.53 | |
| SNR*speech | F(1,330) = 1.45 p = .23 | | ROI*speech | F(1, 1034) = 0.069 p = 0.50 | |

differences between two SNR levels and between the two speech types (Table 2). The greater changes in cortical activity in the DLPFC compared to the AC could be related to effortful speech perception. Consistent with our results, previous neuroimaging studies using fMRI [46] and fNIRS [20] also found different patterns in the LPFC and AC to vocoded and unprocessed speech depending on the attention. Both studies showed greater responses to the vocoded speech compared to the unprocessed speech in the left LPFC when listeners attended to the target speech rather than irrelevant distracters. However, responses in the AC on both sides were not affected by listeners' attention. Taken together, these results suggest that effortful perception of spectrally degraded speech, which requires attentional listening is associated with greater changes in cortical activity in the left LPFC but not AC.

The differences in cortical locations between the current study and the above two studies, i.e., left DLPFC and IFG could be due to the limited spatial resolution of fNIRS compared to fMRI or the differences in recording regions of fNIRS systems. The reported regions could overlap or share the same cognitive functions for effortful speech perception. Alternatively,

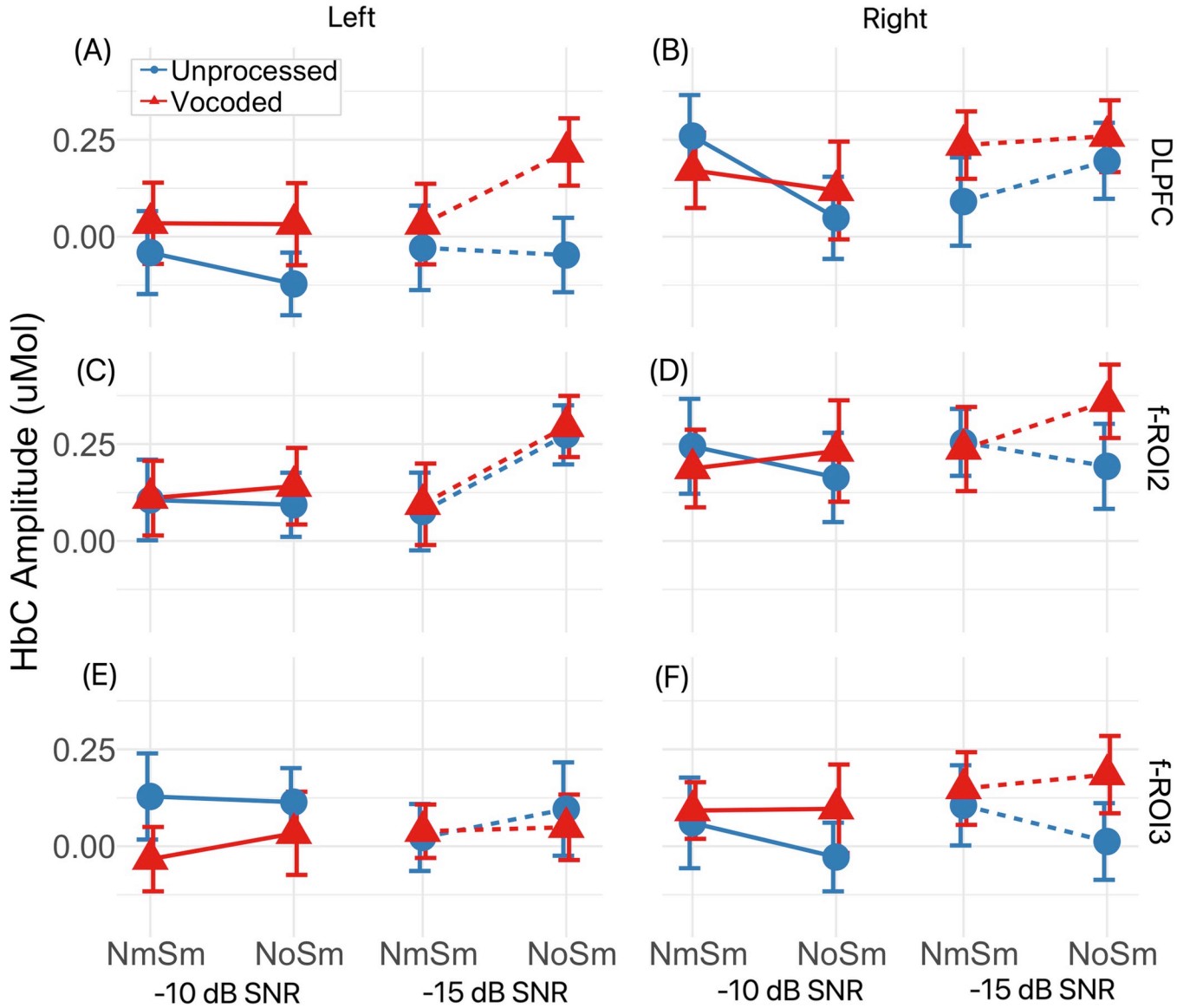

**Fig 7. ΔHbC amplitudes in the three subregions within the LPFC on two hemispheres.**

different sub-regions were involved for effortful speech perception with varying configurations. Hence, we divided the LPFC into three sub-regions, i.e., DLPFC, f-ROI2, and f-ROI3, and attempted to explore fNIRS measures in these regions on both hemispheres. Our results showed a significant effect of speech type with greater ΔHbC amplitudes to vocoded versus unprocessed speech, and significant differences in responses between three subregions, with greater responses in the f-ROI2 compared to the DLPFC and f-ROI3. Further, DLPFC showed greater hemispherical differences compared to f-ROI3. However, there was no significant interaction between speech type * ROI. Our results did not demonstrate significantly different effects of SNR or masker condition between the three sub-regions, either. It is possible that there are no functional differences, and the three subregions in the current study (Fig 1B) still overlap and share common functions in the effortful speech perception with varying stimulus configurations. Each ROI consisted of three 3-cm channels and overlapped on the surface i.e.,

DLPFC (BA, 9 and 10), f-ROI2 (BA, 9 and 46), and f-ROI3 (BA, 45 and 47). As shown in the sensitivity map (Fig 1B), the three regions might share some measures of changes in hemoglobin from the same origins. It is also likely that our data was underpowered due to the small sample, and any potential differences between the three subregions could not be assessed. Further, the configurations, i.e., spectral degradation or binaural unmasking at two SNRs, might be too complicated. Future studies are in need to include larger samples or to focus on no more than two factors concurrently when investigating the LPFC role for binaural unmasking and processing spectrally degraded information.

## Evidence of effect of SNR but not binaural unmasking

Although our data suggest a potential signature of task difficulty in the DLPFC, related to worsening SNR, we found no cortical signature corresponding to binaural unmasking in this region. This finding is somewhat surprising, as our behavioral results showed a significant main effect of masker condition for self-reported task difficulty levels, with greater task difficulty in the monaural compared to the diotic conditions. In the former conditions, both speech and noise would be perceived in the same ear (co-located). Whereas, when target speech is presented to one ear and noise is presented to both ears, if binaural integration for the masker occurs, listeners perceive the speech from the left ear, and the noise is perceived as a single, fused auditory image at the center/front of their head; this separation is known to induce binaural unmasking [69, 70]. Binaural unmasking has been shown to improve speech reception threshold (SRT) in noise by 5–8 dB and higher (even 12–15 dB), depending on the speech materials utilized, because spatial separation of target and maskers improves intelligibility and speech understanding [71–73]. Our behavioral results revealed the effect of binaural unmasking by showing better speech intelligibility and lower self-reported task difficulty levels. We therefore expected some evidence of binaural unmasking in the neural activity. We specifically focused on the left DLPFC, previously demonstrated to boost visuospatial memory capacity in the parietal cortex, and to select the relevant verbal representation in the IFG through top-down control [74, 75]. However, our fNIRS measures did not show a significant main effect of masker condition in the left DLPFC. We considered the potential effect of the differences in the monaural masker noise level, as we reduced the noise level by 3 dB (hence raising the monaural SNR by 3 dB in the $N_oS_m$ conditions to compensate for the otherwise doubling of intensity). When examining the effect of binaural unmasking, by comparing fNIRS measures between $N_oS_m$ and $N_mS_m$ conditions, the 3 dB increase in the monaural SNR in the $N_oS_m$ condition could have diminished binaural unmasking, eliminating the neural correlate in the LPFC. On the other hand, the 3 dB difference might not affect fNIRS measures, as improved SRT related to binaural unmasking could be up to 5–8 dB or even higher when responding to vocoded speech with same-band masker noise at separated versus co-located conditions [76].

We also considered the possibility that the left DLPFC plays a role in the cortical processing of binaural unmasking. For instance, this study [77] investigated the auditory spatial processing in patients who had focal left or right hemisphere damage. Their results demonstrated that right hemispheric damage caused the imprecision of distinguishing sound presented from both hemispaces, whereas left hemispheric damage only caused imprecision in the contralateral hemisphere. Another study [37] investigated cortical activity in the LFCx for target speech with speech masker co-located versus separated from the speech. They found that the right LFCx showed significantly greater responses in the spatially separated versus co-located conditions. Therefore, as a secondary analysis, we examined the effect of varying configurations on the three subregions on the LPFC on both hemispheres. Our fNIRS results demonstrated a

significant main effect of SNR, with greater $\Delta$HbC amplitude at lower (-15 dB) versus higher (-10 dB) SNR (Table 2). We also found a significant interaction between masker condition and SNR, with greater differences in the $\Delta$HbC amplitudes for the LPFC between $N_oS_m$ and $N_mS_m$ conditions at -15 dB SNR compared to at -10 dB SNR. Greater responses to $N_oS_m$ versus $N_mS_m$ are consistent with results from this previous study [37] but opposite to our hypothesis, which proposed that binaural unmasking would reduce the task demand, hence resulting in smaller responses in the $N_oS_m$ conditions. These results suggest that binaural unmasking might affect cortical activity by improving the salience of speech in noise rather than through reducing listening effort. When SNR was lower (-15 dB SNR), i.e., the speech was softer, improving speech salience by separating speech and masker locations better enhanced speech perception compared to when SNR was higher (-10 dB SNR). Further, the differences between studies in the hemisphere effects could be due to different experimental configurations. Indeed, using fNIRS, our study showed a significant effect of SNR in the LPFC on both hemispheres; whereas another fNIRS study [42] found elevated cortical activity in the right DLPFC in response to speech narratives recorded from naturalistic and noisy environment as the SNR decreased.

Surprisingly, no evidence of binaural unmasking was found in the AC, which has been reported as a neural marker of binaural unmasking in both humans [31, 32] and other mammals [29]. Much like in the left DLPFC, the AC faces the issue of limited near-infrared light penetration. In adults, the near-infrared light penetration is about 1.5 cm [78], which could be too shallow for the primary AC, which lies in the deeper superior temporal sulcus. Hence, optodes channels above AC might not be able to detect good neural signal to noise ratios. Alternatively, it could be again due to the differences in experimental configurations and protocols, as some other studies [33, 34] did not demonstrate evidence of binaural unmasking in the AC.

## Non-monotonic responses in the LPFC with task demands

Our results showed a significant and positive correlation between $\Delta$HbC amplitudes in the left DLPFC with self-reported task difficulty levels. However, in the condition that was self-reported as most difficult with the lowest speech intelligibility (vocoded with monaural noise, $VN_mS_m$ at -15 dB SNR), $\Delta$HbC amplitudes did not increase compared to the second most difficult condition (Fig 6, $VN_oS_m$), i.e., non-monotonic.

The non-monotonic response in the LPFC in the current study could be first due to a non-linear relation between cognitive resources exerted and the increase in task demands [23, 46]. For instance, this fNIRS study [23] examined the cortical activity in a group of normal-hearing listeners when responding to speech with degraded spectral information and demonstrated a non-monotonic (U-shaped) pattern of $\Delta$HbO responses in the left IFG versus speech intelligibility (0, 25, 50, 75, and 100% correct). The responses in IFG were lower when speech was not intelligible at all or when it was relatively easy to understand (100% correct). The non-monotonic pattern of changes in cortical activity with increase in task demands, were analogous to the patterns of pupil dilations with varying levels of speech intelligibility [79–82]. For instance, this pupillometry study [80] recorded pupil dilation by presenting speech sentences with varying levels of masking noise. They demonstrated that pupil dilation peaked at an intermediate intelligibility level and decreased when speech intelligibility was either so poor that it was close to floor level or so good that it reached ceiling level.

We also considered the possibility of LPFC responses being modulated by varying task demands related to decision-making and working memory. Although the current study was designed to measure speech perception while recording fNIRS data, it could be argued that the

task involved a decision-making component as well. During the task, while listening to running speech, participants had to detect the color words, hold that information in memory, add up the number and decide if it was even/odd. Listening to degraded versus unprocessed speech, with monaural versus diotic noise, made it harder to recognize the words, thus potentially hindering the memory recall and decision regarding the color. Congruent with theory, greater activity was found in the right LPFC when participants were listening to degraded versus unprocessed speech and making decisions that involved semantic or syllable processing in a PET study [83]. The increased activity in the LPFC to vocoded speech in [83] and in the current study is compatible with the role of the PFC for decision making in the auditory detection tasks, resulting in increased cognitive demands.

To summarize, the non-monotonic pattern of cortical activity in the current study could be due to the high task demands in the most difficult condition, which also resulted in very poor speech intelligibility (Fig 4). It could also be modulated by increased attention and engagement for decision making and greater working memory demands when the task was reported as being more difficult.

## Hemispheric differences are unrelated to handedness

Our results found a significant main effect of hemisphere for ΔHbC amplitudes, with greater response amplitudes on the right hemisphere (Table 2), compared to the left. These results may have occurred because in the current study speech was presented to the left ear alone, while the noise was either presented to the left ear or both ears through insert earphones. Auditory perception involves the AC in both hemispheres where greater contralateral representation is well known to exist [83, 84], i.e., greater responses on the side of the brain opposite to the ear of stimulation than in the ipsilateral side. In the previous two studies that also examined effortful speech perception of degraded speech, no hemisphere difference was found [20, 46]. These inconsistencies might be accounted for by the use of free-field stimuli [20] or diotic stimulation [46] permitting sounds to reach both ears in all conditions.

We considered whether the significant differences in fNIRS responses between the two hemispheres were driven by the handedness of the participants. Handedness has also been reported to affect asymmetric cortical activity involved in speech processing and localization [85, see reviews by 86]. According to the statistics reported in [86], speech processing in 97% of the right-handed participants is left lateralized, and is right lateralized in the remaining 3% of participants. Whereas, in the left-handed participants the ratios shift to 70% and 30%, respectively. The majority of asymmetrical cortical activity is found in the planum temporale and other primary and association auditory cortices [87]. For Broca's area (BA 44 and 45), handedness was found to have affected the asymmetries of the par opercularis (BA 44), with the right-handed participants showing left-hemisphere asymmetry and the left-handed subjects showing right-hemisphere asymmetry [85]. In this fNIRS study [20], a small number of left-handed participants showed increased activity in the right IFG when listening to degraded speech versus unprocessed speech, opposite to what was found in their group of right-handed participants. Though the results were not significant due to the small sample size, they suggested that the laterality of IFG, which showed signs of effortful activity, might be related to the handedness of subjects. However, this theory is insufficient for explaining our results, as 21 out of twenty-three participants were right-handed and the significant activity was found in the right hemisphere.

To summarize the fNIRS findings, the significant results found here in the LPFC could be driven by several factors such as effortful listening for speech perception, attention and task

engagement for decision making, working memory demands, and contralateral stimulation from the left ear. However, the effects did not stem from handedness.

## Conclusion

The current study investigated whether neural signatures for binaural unmasking could be identified by examining cortical activity using fNIRS. Our results demonstrated significant differences between the left DLPFC and the AC, in responses to vocoded versus unprocessed speech, at two SNRs that were 5 dB apart, suggesting that these anatomical areas may play different roles in speech perception, in line with previous findings. Our fNIRS data did not demonstrate evidence of binaural unmasking in the LPFC; however, a significant interaction between SNR and masker condition suggests that binaural unmasking affects cortical activity in the LPFC through improving SRT rather than reducing effort exerted. The result that no significant regional differences existed within the LPFC suggests that these regions might share common cognitive functions in response to effortful speech perception in the current configurations.

## Supporting information

**S1 Data.**
(XLS)

## Acknowledgments

The authors appreciate the time and support from all the research participants. We also thank various colleagues from the Binaural Hearing and Speech Lab for participants recruitment and for suggestions regarding experimental design, including Alan Kan, Emily Burg, Z. Ellen Peng, Shelly Godar, and Tanvi Thakkar.

## Author Contributions

**Conceptualization:** Xin Zhou, Gabriel S. Sobczak, Colette M. McKay, Ruth Y. Litovsky.

**Formal analysis:** Xin Zhou.

**Funding acquisition:** Ruth Y. Litovsky.

**Investigation:** Xin Zhou, Gabriel S. Sobczak.

**Methodology:** Xin Zhou, Gabriel S. Sobczak.

**Project administration:** Xin Zhou.

**Resources:** Ruth Y. Litovsky.

**Supervision:** Ruth Y. Litovsky.

**Visualization:** Colette M. McKay.

**Writing – original draft:** Xin Zhou.

**Writing – review & editing:** Xin Zhou, Gabriel S. Sobczak, Colette M. McKay, Ruth Y. Litovsky.

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
