## [Decision Letter · Decision Letter 0]

8 Mar 2022

PONE-D-21-30090Effects of degraded speech processing and binaural unmasking investigated using functional near-infrared spectroscopy (fNIRS)PLOS ONE

Dear Dr. Zhou,

Thank you for submitting your manuscript to PLOS ONE. After careful consideration, we feel that it has merit but does not fully meet PLOS ONE’s publication criteria as it currently stands. Therefore, we invite you to submit a revised version of the manuscript that addresses the points raised during the review process.

Both reviewers felt that the manuscript is a significant contribution to our understanding of the neural basis of listening effort and spatial (un)masking. The larger concerns stem from questions about whether it is possible with fNIRS to spatially resolve regions of frontal cortex that are analyzed separately (i.e. IFG vs. DLPFC) and whether it is possible at all to measure activity from DMPFC given its depth. I agree with the recommendation of both reviewers that it would be better and technically more correct to lump these regions into "lateral prefrontal cortex" given that the main questions are about AC vs PFC and the fact that these data probably don't permit dissociation of sub-regions of PFC. 

We look forward to receiving your revised manuscript.

Kind regards,

Andrew R Dykstra

Academic Editor

PLOS ONE

2. Please provide additional details regarding participant consent. In the Methods section,

please ensure that you have specified (1) whether consent was informed and (2) what type you obtained

(for instance, written or verbal). If your study included minors, state whether you obtained consent from parents

 or guardians. If the need for consent was waived by the ethics committee, please include this information.

3. Please change "female” or "male" to "woman” or "man" as appropriate, when used as a noun (see for instance https://apastyle.apa.org/style-grammar-guidelines/bias-free-language/gender).

“This study was supported by National Institute on Deafness and Other Communication Disorders (NIH-NIDCD, R01DC003083 to RL), UW-Madison’s Office of the Vice Chancellor for Research, and a Core grant from NIH-NICHD (U54HD090256 to Waisman Center).”

5. Thank you for stating the following in the Funding Section of your manuscript:

“This study was supported by NIH-NIDCD (R01DC003083 to RL), UW-Madison’s Office of the Vice Chancellor for Research, and a Core grant from NIH-NICHD (U54HD090256 to Waisman Center).”

“This study was supported by National Institute on Deafness and Other Communication Disorders (NIH-NIDCD, R01DC003083 to RL), UW-Madison’s Office of the Vice Chancellor for Research, and a Core grant from NIH-NICHD (U54HD090256 to Waisman Center).”

6. Thank you for stating the following in the Competing Interests section:

“I have read the journal's policy. Dr. Litovsky discloses that she is a consultant for Frequency Therapeutics. The other authors have all certified that they have NO affiliations with or involvement in any organization or entity with any financial interest or non-financial interest in the subject matter or materials discussed in this manuscript.”

Reviewers' comments:

Reviewer's Responses to Questions

**Comments to the Author**

1. Is the manuscript technically sound, and do the data support the conclusions?

Reviewer #1: Yes

Reviewer #2: Yes

2. Has the statistical analysis been performed appropriately and rigorously? 

Reviewer #1: Yes

Reviewer #2: Yes

3. Have the authors made all data underlying the findings in their manuscript fully available?

Reviewer #1: Yes

Reviewer #2: Yes

4. Is the manuscript presented in an intelligible fashion and written in standard English?

Reviewer #1: Yes

Reviewer #2: Yes

5. Review Comments to the Author

Reviewer #1: I have read the following submission carefully and consider it to be of good quality, hypothesis driven, and methodologically transparent. The article describes a speech in noise task in which listeners must attend to speech that is either unprocessed or degraded by an 8-channel vocoder. The key measures are from fNIRS centered on primary auditory centers in both hemispheres. The authors tested the hypothesis that two specific areas (left AC and left DLPFC) would carry different information regarding the nature of performing on the task, specifically regarding attention versus speech intelligibility factors during effortful listening. If there was a single flaw in the presentation of the paper, it would be too much consideration of listening effort when setting up the motivation or discussing the results. The authors measured subjective assessments of task difficulty as an indirect measure of listening effort for separate individuals using a different task (counting versus speech recognition), and no objective measure of the participants in the fNIRS test was used. Therefore, in my opinion, the space dedicated to its potential relevance is mostly conjecture and unnecessary for the stated hypotheses (ex. Line 523). That aside, the paper offers additional evidence of the specific roles that primary auditory centers have in speech-in-noise listening, and therefore, it is good contribution to the field.

The following are minor but should be addressed:

1) At various points (ex. Line 175), the authors refer to the poor fNIRS spatial resolution of previous studies (i.e., only 3-4 channels over certain areas). It’s unclear how that relates to the present paper other than maybe from counting dots in Figure 1. There’s a potential spot to mention the difference on Line 549.

2) Initially I flagged the stimulus methods for reducing the level of the masker by 3dB in the diotic condition; however, this was addressed in the discussion as a possible contributing factor to the results. It would be good, however, to provide a source reference for choosing 3 dB as the reduction on or around Line 235. Because this section specifically refers to loudness perception, and BLDELs can exceed 3, even 5 dB, for tonal stimuli, and it is common to have large individual variability, it would be helpful to acknowledge this upfront.

3) I may have missed it, but how accurate were listeners at counting color words and labeling odd/even for the fNIRS task? Were trials treated differently depending on their accuracy?

Others:

1) Title page has alphabet superscripts and numeral affiliations

2) I am not familiar with the in-text citation format that includes the first two of multiple authors (e.g., Line 108), but for consistency, there was at least two instances where the more traditional et al was used instead (line 560 or 636).

3) Typo: “unmaaking” (Line 97)

4) Delete “the” between “in….challenging conditions”

5) Line 228: This IEEE citation was not in the references

6) Line 230-231: I’m not sure what this is in parentheses after ER-2A phones – should be Etymotic?

7) Line 522: move “the” to after “between” (not before).

8) Line 633: change sNR to SNR

9) Line 715: The word “unfortunately” in this context sounds like the authors had a personal stake or interest in the results which I doubt is the case. I suggest removing this transition word.

Reviewer #2: The authors investigated the differential involvement of the auditory cortex (AC) and dorsolateral prefrontal (DLPFC) cortex in different listening conditions that manipulate the spatial configuration of noise (monotic vs diotic inducing spatial release from masking), signal-to-noise ratio as well as speech processing (using a vocoder simulation or left unprocessed). They found that DLPFC and AC showed different patterns of activities for SNR and vocoded speech manipulation but did not find evidence of spatial unmasking changing activities in these areas. Furthermore, they found a correlation with perceived task difficulty and neural activity in the left DLPFC. Overall, this is a well written manuscript that uses sophisticated fNIRS analyses and appropriate statistical approaches. Nevertheless, I have the following concerns:

Major:

1. It is unclear why the secondary question is there in the first place when the spatial resolution of fNIRS is clearly not sufficient to resolve these regions (as evident in the anatomical overlaps shown in Figure 1B). Perhaps the authors wanted to bring in the different literature to talk about listening effort and speech processing and thus needed to mention DLPFC and Inferior Frontal Gyrus (IFG) as potential regions of interest. Personally, I think it would just be cleaner to say that you have optodes over the lateral prefrontal cortex (and not make a delineation of dorsal vs ventral) and that the effects you see can be attributed to DLPFC and / or IFG. I think the authors already made the case in the discussion that ll545-7 "The lack of difference could also be due to the poor spatial resolution of fNIRS and measures from the three subregions could have still overlapped" so why present them in the first place?

2. I'm also doubtful that the medial prefrontal cortex is also part of the ROI because of how deep that structure is. As a reader, I also want to be convinced the the medial cortex can be captured in these optodes and only then bring in the literature regarding medial PFC in working memory, decision-making in your discussion.

3. p<0.09 should never be cast as "marginally non significant." Please remove this in the abstract and elsewhere in the manuscript.

Minor:

1. In-text citation looks funny when the authors are mentioned (e.g., Ln 108, it should be Hughes et al and not Hughes, Rowe and there are many instances throughout the MS that needs to be fixed).

2. Fig 5 is missing SNR labels

3. Ln 523: why Left AC here and not just AC?

4. Ln 633 Change sNR to SNR

6. PLOS authors have the option to publish the peer review history of their article (what does this mean?). If published, this will include your full peer review and any attached files.

Reviewer #1: **Yes: **Erol Ozmeral

Reviewer #2: **Yes: **Adrian KC Lee

---

## [Author Response · Author response to Decision Letter 0]

29 Mar 2022

Dear Editor, 

We want to thank you and two reviewers for providing us with excellent feedback to further improve the quality of this manuscript. We have worked to clarify the two major concerns, i.e., too much discussion about listening effort from Reviewer #1, and fNIRS’ poor spatial resolution from Reviewer #2. We provide a point-by-point response below. We look forward to hearing back from you about the suitability of the paper for publication. 

Editor’s comment

Both reviewers felt that the manuscript is a significant contribution to our understanding of the neural basis of listening effort and spatial (un)masking. The larger concerns stem from questions about whether it is possible with fNIRS to spatially resolve regions of frontal cortex that are analyzed separately (i.e. IFG vs. DLPFC) and whether it is possible at all to measure activity from DMPFC given its depth. I agree with the recommendation of both reviewers that it would be better and technically more correct to lump these regions into "lateral prefrontal cortex" given that the main questions are about AC vs PFC and the fact that these data probably don't permit dissociation of sub-regions of PFC. 

Response: We appreciate the concern that the spatial resolution of fNIRS might not permit a distinction of DLPFC from IFG. Because we are interested in the questions raised in the study regarding varying configurations on cortical activity in the sub-regions within LPFC, in the revision, we did not label the two regions as DMPFC or IFG. Instead, we labelled them as f-ROI2 and f-ROI3, and spelled out their correspondence with Broadman areas on the surface. Though our results did not find functional differences across the three sub-regions in varying configurations, we still showed significant differences across ROIs. We therefore elaborated on our interpretations of the results. Please see our detailed responses to Reviewer #2 below. 

Comments from and responses to Reviewer #1

Reviewer #1: I have read the following submission carefully and consider it to be of good quality, hypothesis driven, and methodologically transparent. The article describes a speech in noise task in which listeners must attend to speech that is either unprocessed or degraded by an 8-channel vocoder. The key measures are from fNIRS centered on primary auditory centers in both hemispheres. The authors tested the hypothesis that two specific areas (left AC and left DLPFC) would carry different information regarding the nature of performing on the task, specifically regarding attention versus speech intelligibility factors during effortful listening. If there was a single flaw in the presentation of the paper, it would be too much consideration of listening effort when setting up the motivation or discussing the results. The authors measured subjective assessments of task difficulty as an indirect measure of listening effort for separate individuals using a different task (counting versus speech recognition), and no objective measure of the participants in the fNIRS test was used. Therefore, in my opinion, the space dedicated to its potential relevance is mostly conjecture and unnecessary for the stated hypotheses (ex. Line 523). That aside, the paper offers additional evidence of the specific roles that primary auditory centers have in speech-in-noise listening, and therefore, it is good contribution to the field.

Response: We appreciate this comment about over discussing listening effort. In the revision, we deleted the sentence in the introduction ‘This result was consistent with the idea that task demands and motivation are related (3, 42), where listening effort increases with task difficulty but decreases again due to lack of motivation when the task becomes too difficult.’ 

In the discussion, we cut down listening effort and just focused on the measure of task demands. We reorganized the subsection of ‘Non-monotonic responses in the LPFC with task demands’ (lines 645-681), and discussed the non-monotonic pattern of cortical activity in the current study could be due to the high task demands in the most difficult condition, which also resulted in very poor speech intelligibility; this pattern could also be modulated by decision making and working memory that are involved when performing the task. 

The following are minor but should be addressed:

1) At various points (ex. Line 175), the authors refer to the poor fNIRS spatial resolution of previous studies (i.e., only 3-4 channels over certain areas). It’s unclear how that relates to the present paper other than maybe from counting dots in Figure 1. There’s a potential spot to mention the difference on Line 549.

Response: We appreciate this comment. In the revision (lines 257-260), we included the sentence that ‘Each sub-region consisted of three channels (Fig. 1B). The DLPFC corresponded to Broadman area (BA, 9 and 10) on the surface, f-ROI2 corresponded to BA 9 and 46, and f-ROI3 corresponded to BA45 and BA47, which likely covers the IFG.’

We appreciate the suggestion about elaborating on the differences across the three subregions. In the discussion (lines 574-579), we included the sentences ‘It is possible that there are no functional differences, and the three subregions in the current study (Fig. 1B) still overlap and share common functions in the effortful speech perception with varying stimulus configurations. Each ROI consisted of three 3-cm channels and overlapped on the surface i.e., DLPFC (BA, 9 and 10), f-ROI2 (BA, 9 and 46), and f-ROI3 (BA, 45 and 47). As shown in the sensitivity map (Fig. 1B), the three regions might share some measures of changes in hemoglobin from the same origins.’ 

2) Initially I flagged the stimulus methods for reducing the level of the masker by 3dB in the diotic condition; however, this was addressed in the discussion as a possible contributing factor to the results. It would be good, however, to provide a source reference for choosing 3 dB as the reduction on or around Line 235. Because this section specifically refers to loudness perception, and BLDELs can exceed 3, even 5 dB, for tonal stimuli, and it is common to have large individual variability, it would be helpful to acknowledge this upfront.

Response: We appreciate this comment. We agree BLDELs can be more than 3 dB and vary across stimuli. The 3 dB adjustment was planned to equalize the sound pressure level in two conditions, rather than the perception of loudness. We rephrased this sentence as ‘The 3-dBA reduction in the NoSm condition was introduced to compensate for the otherwise doubling of intensity, so that the sound pressure level would be equalized between NmSm and NoSm conditions’. 

3) I may have missed it, but how accurate were listeners at counting color words and labeling odd/even for the fNIRS task? Were trials treated differently depending on their accuracy?

Response: We appreciate this comment. In the revision we included the sentence that ‘All the blocks were included, regardless of participants’ accuracies in pushing a mouse button to indicate hearing an even or odd number of color words, except for individual blocks that had values 2.5 standard deviations above or below the mean of the group.’

Others:

1) Title page has alphabet superscripts and numeral affiliations

Response: fixed! 

2) I am not familiar with the in-text citation format that includes the first two of multiple authors (e.g., Line 108), but for consistency, there was at least two instances where the more traditional et al was used instead (line 560 or 636).

Response: We appreciate this comment and revised the citation format through the text to be consistent. 

3) Typo: “unmaaking” (Line 97)

Response: fixed! 

4) Delete “the” between “in….challenging conditions”

Response: fixed! 

5) Line 228: This IEEE citation was not in the references

Response: We appreciate the comment and included the citation in the revision. 

6) Line 230-231: I’m not sure what this is in parentheses after ER-2A phones – should be Etymotic?

Response: Thank you for the comment. We revised the citation for this product. 

7) Line 522: move “the” to after “between” (not before).

Response: fixed! 

8) Line 633: change sNR to SNR

Response: fixed! 

9) Line 715: The word “unfortunately” in this context sounds like the authors had a personal stake or interest in the results which I doubt is the case. I suggest removing this transition word.

Response: We deleted this transition word, and also moved this sentence up to for a better flow in Conclusions. Please see lines 720-725. 

Reviewer #2: The authors investigated the differential involvement of the auditory cortex (AC) and dorsolateral prefrontal (DLPFC) cortex in different listening conditions that manipulate the spatial configuration of noise (monotic vs diotic inducing spatial release from masking), signal-to-noise ratio as well as speech processing (using a vocoder simulation or left unprocessed). They found that DLPFC and AC showed different patterns of activities for SNR and vocoded speech manipulation but did not find evidence of spatial unmasking changing activities in these areas. Furthermore, they found a correlation with perceived task difficulty and neural activity in the left DLPFC. Overall, this is a well written manuscript that uses sophisticated fNIRS analyses and appropriate statistical approaches. Nevertheless, I have the following concerns:

Major:

1. It is unclear why the secondary question is there in the first place when the spatial resolution of fNIRS is clearly not sufficient to resolve these regions (as evident in the anatomical overlaps shown in Figure 1B). Perhaps the authors wanted to bring in the different literature to talk about listening effort and speech processing and thus needed to mention DLPFC and Inferior Frontal Gyrus (IFG) as potential regions of interest. Personally, I think it would just be cleaner to say that you have optodes over the lateral prefrontal cortex (and not make a delineation of dorsal vs ventral) and that the effects you see can be attributed to DLPFC and / or IFG. I think the authors already made the case in the discussion that ll545-7 "The lack of difference could also be due to the poor spatial resolution of fNIRS and measures from the three subregions could have still overlapped" so why present them in the first place?

Response: We appreciate this comment. As previous studies reported different regions of interest within the LPFC, we asked an exploratory research question whether there are functional differences across sub-regions within the LPFC in varying configurations. We acknowledge that fNIRS has limited spatial resolution hence unable to measure response from DMPFC. In the revision, we made changes below to help clarify our rationale and interpretation of results. 

• In the introduction (lines 67-68), we phrased the secondary aim of this study as ‘Our secondary question was whether there were functional differences across sub-regions within the LPFC in varying stimulus configurations.’ 

• In the introduction (lines 167-173), to motivate the rationale of exploring regional differences, we included ‘The differences in regions involved could be partially due to the varying number of optodes employed in different fNIRS montages across experiments which have impacted the size of recording region (surface area). For instance, 4 channels on the LFCx in Zhang et al. (37) permitted a greater recording area compared with the 3 channels on the IFG in Wijayasiri et al. (20). Without a good coverage of the frontal area, it is difficult to parse out whether LPFC subregions overlap and share common functions, and whether the regions reported in the fNIRS studies (20, 23, 37) overlap with the regions reported in fMRI studies (36, 46, 47)’. 

• As suggested by the other reviewer, we cut down our discussions of listening effort as we did not have objective measures of effort from the fNIRS session, nor was listening effort most related to our hypotheses. Please see our response above. 

• In the discussion (lines 576-584), to interpret the results regarding the three sub-regions, we add the sentences ‘Each ROI consisted of three 3-cm channels and overlapped on the surface i.e., DLPFC (BA, 9 and 10), f-ROI2 (BA, 9 and 46), and f-ROI3 (BA, 45 and 47). As shown in the sensitivity map (Fig. 1B), the three regions might share some measures of changes in hemoglobin from the same origins. It is also likely that our data was underpowered due to the small sample, and any potential differences between the three subregions could not be assessed. Further, the configurations, i.e., spectral degradation or binaural unmasking at two SNRs, might be too complicated. Future studies are in need to include larger samples or to focus on no more than two factors concurrently when investigating the LPFC role for binaural unmasking and processing spectrally degraded information.’

2. I'm also doubtful that the medial prefrontal cortex is also part of the ROI because of how deep that structure is. As a reader, I also want to be convinced the the medial cortex can be captured in these optodes and only then bring in the literature regarding medial PFC in working memory, decision-making in your discussion.

Response: We appreciate this comment. In the revision, we did not label DMPFC and IFG, instead, we used ‘two adjacent regions of interests within LPFC’ and noted as f-ROI2 and f-ROI3’. We also agreed that we possibly stretched a bit far with our discussion regarding DMPFC activity related to decision-making and working memory. In the revision, we cut down this discussion about VMPFC, reorganized our discussion, and ‘considered the possibility of LPFC responses being modulated by varying task demands related to decision-making and working memory.’ Please see lines 664-681. 

3. p<0.09 should never be cast as "marginally non significant." Please remove this in the abstract and elsewhere in the manuscript.

Response: We appreciate this comment and described p=0.09 as non-significant through the text in the revision. 

Minor:

1. In-text citation looks funny when the authors are mentioned (e.g., Ln 108, it should be Hughes et al and not Hughes, Rowe and there are many instances throughout the MS that needs to be fixed).

Response: We appreciate this comment and revised the citation format through the text to be consistent. 

2. Fig 5 is missing SNR labels

Response: fixed! 

3. Ln 523: why Left AC here and not just AC?

Response: fixed! 

4. Ln 633 Change sNR to SNR

Response: fixed!

---

## [Decision Letter · Decision Letter 1]

12 Apr 2022

Effects of degraded speech processing and binaural unmasking investigated using functional near-infrared spectroscopy (fNIRS)

PONE-D-21-30090R1

Dear Dr. Zhou,

We’re pleased to inform you that your manuscript has been judged scientifically suitable for publication and will be formally accepted for publication once it meets all outstanding technical requirements.

Kind regards,

Andrew R Dykstra

Academic Editor

PLOS ONE

Additional Editor Comments (optional):

Reviewers' comments:

Reviewer's Responses to Questions

**Comments to the Author**

1. If the authors have adequately addressed your comments raised in a previous round of review and you feel that this manuscript is now acceptable for publication, you may indicate that here to bypass the “Comments to the Author” section, enter your conflict of interest statement in the “Confidential to Editor” section, and submit your "Accept" recommendation.

Reviewer #1: All comments have been addressed

Reviewer #2: All comments have been addressed

2. Is the manuscript technically sound, and do the data support the conclusions?

Reviewer #1: Yes

Reviewer #2: Yes

3. Has the statistical analysis been performed appropriately and rigorously? 

Reviewer #1: Yes

Reviewer #2: Yes

4. Have the authors made all data underlying the findings in their manuscript fully available?

Reviewer #1: Yes

Reviewer #2: Yes

5. Is the manuscript presented in an intelligible fashion and written in standard English?

Reviewer #1: Yes

Reviewer #2: Yes

6. Review Comments to the Author

Reviewer #1: All comments have been addressed. Please check new text for typos, e.g.:

Line 360: inlcuded -> included

Line 725: exisited -> existed

Reviewer #2: The authors have addressed all the comments except I urge them to consider these final recommendations: the DLPFC region in fig 1B is in general not different from f-ROI2 and f-ROI3. I suggest labeling it as f-ROI1 instead. Also, I suggest the following wordings on ll 260-262: The f-ROI1 most likely encompasses Broadman area (BA, 9 and 10), f-ROI2, BA 9 and 46, and f-ROI3, BA45 and BA47, which also likely covers the IFG

7. PLOS authors have the option to publish the peer review history of their article (what does this mean?). If published, this will include your full peer review and any attached files.

Reviewer #1: **Yes: **Erol J. Ozmeral

Reviewer #2: **Yes: **Adrian KC Lee, ScD

---

## [Editor Report · Acceptance letter]

14 Apr 2022

PONE-D-21-30090R1 

Effects of degraded speech processing and binaural unmasking investigated using functional near-infrared spectroscopy (fNIRS) 

Dear Dr. Zhou:

I'm pleased to inform you that your manuscript has been deemed suitable for publication in PLOS ONE. Congratulations! Your manuscript is now with our production department. 

Kind regards, 

on behalf of

Dr. Andrew R Dykstra 

Academic Editor

PLOS ONE